# Translation, Cross-Cultural Adaptation, and Validation of the Japanese Version of the Patient Education Materials Assessment Tool (PEMAT)

**DOI:** 10.3390/ijerph192315763

**Published:** 2022-11-26

**Authors:** Emi Furukawa, Tsuyoshi Okuhara, Hiroko Okada, Ritsuko Shirabe, Rie Yokota, Reina Iye, Takahiro Kiuchi

**Affiliations:** 1Department of Health Communication, Graduate School of Medicine, The University of Tokyo, Tokyo 113-8655, Japan; 2Department of Health Communication, School of Public Health, The University of Tokyo, Tokyo 113-8655, Japan

**Keywords:** patient education, education materials, health communication, health literacy, assessment, measurement, readability

## Abstract

Background: The Patient Education Materials Assessment Tool (PEMAT) systematically evaluates the understandability and actionability of patient education materials. This study aimed to develop a Japanese version of PEMAT and verify its reliability and validity. Methods: After assessing content validation, experts scored healthcare-related leaflets and videos according to PEMAT to verify inter-rater reliability. In validation testing with laypeople, the high-scoring material group (*n* = 800) was presented with materials that received high ratings on PEMAT, and the low-scoring material group (*n* = 799) with materials that received low ratings. Both groups responded to the understandability and actionability of the materials and perceived self-efficacy for the recommended actions. Results: The Japanese version of PEMAT showed strong inter-rater reliability (PEMAT-P: % agreement = 87.3, Gwet’s AC1 = 0.83. PEMAT-A/V: % agreement = 85.7, Gwet’s AC1 = 0.80). The high-scoring material group had significantly higher scores for understandability and actionability than the low-scoring material group (PEMAT-P: understandability 6.53 vs. 5.96, *p* < 0.001; actionability 6.04 vs. 5.49, *p* < 0.001; PEMAT-A/V: understandability 7.65 vs. 6.76, *p* < 0.001; actionability 7.40 vs. 6.36, *p* < 0.001). Perceived self-efficacy increased more in the high-scoring material group than in the low-scoring material group. Conclusions: Our study showed that materials rated highly on Japanese version of PEMAT were also easy for laypeople to understand and action.

## 1. Introduction

A variety of patient education materials, including pamphlets, web pages, videos, and smartphone apps, support patients in the medical field, helping them understand their conditions, make decisions, and communicate with their health care providers. However, studies show that patient education materials are often poorly understood by patients, especially those with limited health literacy [1,2]. Inadequate health literacy is associated with more limited disease control, medical adherence, and patient outcomes [1,3]. Therefore, patient-friendly materials, regardless of the reader’s health literacy, are essential in improving health outcomes for patients.

To address the situation, we focused on the Patient Education Materials Assessment Tool (PEMAT), a reliable and valid instrument developed by the Agency for Healthcare Research and Quality (AHRQ) to evaluate the understandability and actionability of patient education materials [4,5]. Understandability refers to the extent to which consumers of diverse backgrounds can process and explain key messages [4]. Actionability refers to the degree to which consumers of diverse backgrounds and varying levels of health literacy can identify what actions they should take to improve their health, based on the presented information [4]. PEMAT is divided into two parts: PEMAT-P, a scale for printable materials (brochures and PDFs), and PEMAT-A/V, a scale for audiovisual materials (videos and multimedia materials including smartphone apps). The scores are calculated by taking the sum of the points, dividing by the total possible points, and multiplying by 100 to obtain a percentage. The developers have set the cutoff value for understandability and actionability at 70%**.**

Studies have used PEMAT to identify issues with patient materials. For example, Yiu et al. evaluated web-based education materials for patients taking non-vitamin K oral anticoagulants. The study revealed the need to include more summaries of information, visual aids, and tangible tools such as checklists [6]. PEMAT is also reported to be useful in developing or improving patient education materials. Jamil et al. developed an integrated diabetes-periodontitis nutrition and health education module using PEMAT [7].

PEMAT has been translated into Malay [8,9] and Korean [10], but a Japanese version has not yet been developed. Lee et al. created a cardiovascular disease-prevention material for Korean immigrants based on the Korean version of PEMAT and found that they could improve the understandability and actionability of the material [10]. Therefore, we believe it is essential for PEMAT to be deployed in multiple languages and for findings on understandability and actionability to be accumulated. Furthermore, to our knowledge, there are no validated tools to assess whether the material is understandable and actionable in Japanese. Therefore, our study aimed to translate and cross-culturally adapt the PEMAT into Japanese and verify its reliability and validity.

## 2. Materials and Methods

The development of the Japanese version of the PEMAT consists of five steps. We translated the PEMAT in Step 1, examined content validity in Step 2, examined inter-rater reliability in Step 3, examined convergent validity in Step 4, and examined predictive validity in Step 5.

### 2.1. Stage 1: Translation of PEMAT into Japanese

We translated the PEMAT questionnaire and user’s guide with the permission of AHRQ. The translation and cross-cultural adaptation process were carried out according to the International Society for Pharmacoeconomics and Outcomes Research (ISPOR) Task Force [11] and Guidelines for the Process of Cross-Cultural Adaptation of Self-Report Measures [12]. Two translators whose native language is Japanese independently translated the original PEMAT into Japanese (T1 and T2). A health communication researcher (TO) and a physician (EF) reviewed T1 and T2, integrating them into the Japanese forward-translated version (T-12). Two translators whose native language is English independently back-translated T-12 into English (BT1 and BT2). The back-translators were neither aware of the concept of PEMAT nor involved in the forward translation. After making a comparison and integration of BT1 and BT2, the expert committee, including a health communication researcher (TO) and three medical professionals (physician (EF) and nurses (HU and HO)), created a final Japanese version of PEMAT.

### 2.2. Stage 2: Assessment of Content Validity by the Expert Panel

At a panel meeting of the experts, we determined whether the items apply to the Japanese cultural background, and whether the content is appropriate. The original PEMAT is intended for use by health professionals, health librarians, and other professionals who provide health and medical information to patients and the general public [4,13]. Following this, our study recruited experts with the attributes of potential PEMAT users for the content validity evaluation. The expert panel consisted of twelve members: medical specialists with clinical experience (two doctors, one nurse, one pharmacist, and two dieticians); non-medical specialists (two patient advocates, one editor with experience in developing health and medical materials, two media professionals, and one health communication researcher with a background in education.) At the meeting, content validity was assessed based on: (1) relevance (whether the items were relevant to the constructs of interest in the particular population and context of use), (2) comprehensiveness (whether essential aspects of the construct were missing), and (3) comprehensibility (whether the items were understood by the raters as intended). When there were discrepancies among the panel of experts, we consulted with a third person (TK) for reconciliation. If issues or questions were not resolved at the meeting, we asked the developer of the original PEMAT (CB) for feedback.

### 2.3. Stage 3: Determining the Reliability of the Instrument

We tested the inter-rater reliability of the PEMAT using patient education materials written in Japanese. In this study, we selected materials on the primary prevention of common diseases for use by non-healthcare professionals to reduce the variability of materials outside the scope of what the PEMAT aims to measure. This was because if these materials are intended for patients with a particular disease, the message may differ significantly across different materials depending on the medical condition and associated complications. The eligibility criteria for the materials were as follows: (1) developed by academic societies, government offices, or non-profit organizations; (2) including any of the nine topics presented in Health Japan 21 (2nd edition) [13] such as nutrition and dietary habits, physical activities and exercise, rest and mental health, smoking, alcohol, dental health, diabetes, cardiovascular disease, and cancer; and (3) materials that could be downloaded for free from the Internet. We searched via the two most popular search engines in Japan, Google Japan [14] and Yahoo! Japan [15]. The search terms in Japanese were ‘topic’ (where ‘topic’ was any of the nine topics in Health Japan 21) AND ‘pamphlet’ OR ‘leaflet’ OR ‘video’ OR ‘patients’ OR ‘explanation.’ We then selected the first 100 written materials and the first 50 audiovisual materials from the search results.

The evaluators for this study comprised four experts: two physicians with more than 5 years of clinical experience (EF and RS); one dietitian without clinical experience (RI); and one health communication specialist with an educational background (RY). Two rounds of reliability testing were performed because of the low reliability found in the first round. In each round, the evaluators followed the guidance on the question items and evaluation methods using the Japanese version of the PEMAT User’s Guide before evaluating the material. For the 100 PEMAT-P materials, the first 50 were evaluated by EF and RS and the second 50 by EF and RI. In addition, EF and RY evaluated 50 videos for PEMAT-A/V reliability verification. In the second round of reliability testing, the evaluators switched materials to ensure they did not evaluate the same materials as in the first round. The first 50 of the PEMAT-P materials were evaluated by EF and RS and the second 50 by EF and RI. In the second round of the PEMAT-A/V reliability testing, EF and RY evaluated 50 videos. Each material was assessed only once in the second round of reliability assessments reported in this study. Two evaluators independently judged the material and calculated the overall PEMAT score as a percentage. For all materials, the average scores from the two evaluation results were calculated. The materials with the highest and lowest average scores were selected as the high- and low-scoring materials to be presented to the general public, respectively.

### 2.4. Stage 4: Testing Convergent Validity with Readability Scores

We evaluated the materials included in Stage 3 with a readability scale to test convergent validity. EF proposed the text-readability measurement system “jReadability” [16,17], which is a web system for automatically evaluating the readability of Japanese text, as the developers of the original PEMAT recommended using readability evaluation tools to evaluate the readability of printed materials in addition to the PEMAT [18]. It has been demonstrated that the jReadability formula can predict the difficulty of a text with a high degree of accuracy. Furthermore, it has been shown that differences in readability can be detected even when analyzing data other than those used to create the readability formulas (the Japanese Language Proficiency Test) [19]. To assess readability, EF manually retrieved the text from the printable materials and transcribed the audio from the audiovisual materials. The text from the materials used in Stage 3 was then pasted to Microsoft Word, and any formatting elements that may interfere with the readability assessment (e.g., headings, symbols, author information, and references) were removed. The plain text from each material was assessed using the jReadability online readability calculator. This validated measure calculates readability based on the average length of sentences, the difficulty level of words, and the proportion of grammatical parts of speech and types of characters per sentence. Scores range from 0.5 to 6.4, and a high score indicates that the text is relatively easy to read. Scores of 5.5–6.4 indicate the text is very easy to read; 4.5–5.4 indicate easy; 3.5–4.4 is a neutral evaluation; 2.5–3.4 indicate the text is a little difficult to read; 1.5–2.4 indicate difficult; and 0.5–1.4 indicate that it is very difficult.

### 2.5. Stage 5: Assessment of Predictive Validity by Testing with the General Public

In this stage, we conducted an online survey to determine whether non-experts found the material with high/low PEMAT scores (from the expert evaluation in stage 3) easy/difficult to understand and take action from. The online survey consisted of two studies to test the validity of the PEMAT-P and PEMAT-A/V, one with a leaflet presentation and the other with a video presentation.

#### 2.5.1. Participants

Study participants were recruited from registered monitors of an online survey company (Rakuten Insight). The survey company reported that approximately 2.2 million active monitors (who have logged into their registered accounts within 12 months) registered in the panel as of September 2022 [20]. Men and women who use Japanese as a native language were eligible to participate in the study. We solicited the monitors who met the age criteria described below to participate in the survey via email or push notification via the survey company and conducted a screening survey of all who agreed to participate. In the PEMAT-P study, participants aged from 18 to 69 years were included, and in the PEMAT-A/V part, participants aged from 60 to 79 years were included. This is because the age groups targeted by the materials used for intervention in PEMAT-P and A/V were different, as described below. Participants were excluded in the screening section if they had experience in health care or were restricted from practicing the action recommended in the materials due to illness or injury.

Participants were randomized into two groups using a central computerized random allocation system of the survey company. One group (high-scoring material group) viewed the material that was highly scored by experts in stage 3, while the other (low-scoring material group) viewed the low-scoring material. Participants were not aware of which group they were assigned. We asked participants about the content of the material to see if the participants had viewed the material properly. We also adopted a trap question that requires reading the question carefully. Participants who answered these questions incorrectly were excluded. We stopped recruiting participants when the number of valid responses reached the sample size.

#### 2.5.2. Materials

For testing PEMAT-P, participants viewed leaflets that promote healthy eating habits. The PEMAT-P score of the leaflets was 100% for the high-scoring material group and 69.7% for the low-scoring material group. When testing PEMAT-A/V, we used videos on the topic of locomotive syndrome prevention for the elderly. Locomotive syndrome occurs in conditions with a high risk of motor function decline due to locomotive organ impairment [21]. The overall PEMAT-A/V score of each video was 85.4% (intervention group) and 25.0% (control group), respectively.

#### 2.5.3. Measures

The survey company provided participants’ gender and age, and participants responded to questions about their educational background, annual family income, occupation, marital status, and self-perceived health. Participants also answered questions about their baseline health literacy and perceived self-efficacy. Measuring the change in behavior before and after viewing the materials as an outcome would be ideal to verify whether materials rated highly on the PEMAT are more likely to support participants to take action. However, participants may find it difficult to take action (e.g., cooking healthy meals, going out for exercise) immediately after viewing the materials. It is also not feasible to measure behavioral implementation in an online survey. Bandura stated that, for behavior changes, it is vital to increase self-efficacy, which means confidence in carrying out the behavior and overcoming temptations that prevent change [22,23]. Self-efficacy is likely to change over a shorter period and improve as the stage of behavioral change progresses. We measured participants’ self-efficacy to examine the predictive validity, hypothesizing the understandability and actionability scores assessed on the PEMAT predicted self-efficacy. Health literacy was measured using the 14-item health literacy scale for Japanese adults (HLS-14) [19]. Self-efficacy was measured by the Self-Efficacy Scale for Positive Eating Behavior [20] for PEMAT-P and the Home-Exercise Barrier Self-Efficacy Scale [24] for PEMAT-A/V.

After responding to these questions, participants viewed the relevant materials. They then rated how easy the material was to understand or take action from, on a scale from 1 to 10. They also responded to eight selected items in PEMAT (items 1, 4, 8, 9, 11, 17, 19, and 21) (see Table 1 and Table 2). These items were asked in both the PEMAT-P and PEMAT-A/V studies and were relevant for all the presented materials. At the end of the survey, participants responded about their self-efficacy immediately after the intervention on a scale from 1 to 10. The participants scored the items as 1 if they completely disagreed with the content of the item and 10 if they completely agreed with it.

### 2.6. Statistical Analysis

Inter-rater reliability was used to assess the external consistency of the PEMAT using percentage agreement and Fleiss’ kappa for two evaluators. Fleiss’ kappa is an extension of the more commonly reported Cohen’s kappa. However, Cohen’s kappa requires that all materials are evaluated by the same evaluator, whereas Fleiss’ kappa allows for two evaluators, chosen from a pool of potential evaluators [25]. We also calculated Gwet’s AC1 [26] when low kappa values were observed despite a high percentage of agreement [27]. In addition, we calculated the IRR for the summary scales for understandability and actionability. As understandability and actionability scores are quantitative variables, we used Shrout and Fleiss’ intraclass coefficient (ICC) to determine the reliability [28]. Inter-rater agreement was deemed poor (0), slight (0.01–0.20), fair (0.21–0.40), moderate (0.41–0.60), substantial (0.61–0.80), or almost perfect (0.81–1.0) [29]. Pearson’s correlation coefficient was used to determine whether there was a correlation between the PEMAT understandability scores and jReadability scores.

As the questionnaire survey for the general public in this study was designed to measure the accuracy of the scale, we believe that it was crucial to assess the validity of the tool by including only those who correctly answered all questionnaire items. Therefore, we designed Stage 5 to conduct a per-protocol analysis. Sample size calculation was performed based on an effect size of 0.2 (Cohen’s d) [22], a significance level of 0.05, and a power of 0.8. It was estimated that 394 participants per group were required. Differences between the control and intervention groups were evaluated using the two-sample *t*-test for age and the chi-square test or Fisher’s exact test for sex, educational background, occupation, annual household income, marital status, and self-perceived health. Welch’s *t*-test was used to compare understandability, actionability, and perceived self-efficacy between the two groups.

All *p*-values were two-sided, and *p* < 0.05 was considered statistically significant. All analyses were conducted with R version 4.0.3 (10 October 2020).

## 3. Results

### 3.1. Content Validation

At the expert panel meeting, content validity was assessed based on relevance, comprehensiveness, and comprehensibility. The comments and revisions made by the expert panel are shown in Appendix A. All items were deemed necessary by the expert panel, except item number 5. Consequently, the final Japanese PEMAT consists of 25 items and two scales, including understandability (18 items) and actionability (7 items). Details of the text for each item are presented in Table 1 and Table 2.

**Table 1 ijerph-19-15763-t001:** Item-level reliability and validity results for items in the Japanese PEMAT-P.

Item #	Item	Score Results (Sum of Two Raters)						Inter-Rater Reliability									
		1	(%)	0	(%)	N/A	(%)	% Agree	Fleiss’s κ	(95% CI)		Gwet’s AC1	(95% CI)		ICC	(95% CI)	
**UNDERSTANDABILITY**								**88.2**	**0.71**	**0.66**	**0.76**	**0.95**	**0.94**	**0.96**	**0.76**	**0.65**	**0.84**
TOPIC: CONTENT						-	-										
1	The material makes its purpose completely evident from the beginning	117	73.1	43	26.9	-	-	88.8	0.714	0.49	0.93	0.81	0.69	0.94			
2	The material does not include information or content that distracts from its purpose	148	92.5	12	7.5	-	-	90.0	0.279	0.06	0.50	0.88	0.80	0.97			
TOPIC: WORD CHOICE & STYLE																	
3	The material uses common, everyday language	111	69.4	49	30.6	-	-	83.8	0.62	0.40	0.84	0.72	0.56	0.87			
4	When used, medical terms are clearly defined	96	60.0	64	40.0	-	-	92.5	0.84	0.62	1.06	0.86	0.74	0.97			
TOPIC: USE OF NUMBERS																	
5	Numbers appearing in the material are clear and easy to understand	96	60.0	58	36.3	6	3.8	87.5	0.75	0.56	0.95	0.94	0.90	0.98			
6	The material does not expect the user to perform calculations	122	76.3	38	23.8	-	-	92.5	0.79	0.57	1.01	0.88	0.79	0.98			
TOPIC: ORGANIZATION																	
7	The material breaks or “chunks” information into short sections	153	95.6	7	4.4	0	0.0	96.2	0.55	0.33	0.77	0.99	0.98	1.00			
8	The material’s sections have informative headers	152	95.0	8	5.0	0	0.0	95.0	0.47	0.25	0.69	0.99	0.97	1.00			
9	The material presents information in a logical sequence	143	89.4	17	10.6	-	-	86.2	0.28	0.06	0.50	0.83	0.72	0.94			
10	The material provides a summary	40	25.0	120	75.0	0	0.0	85.0	0.60	0.38	0.82	0.94	0.91	0.97			
TOPIC: LAYOUT & DESIGN																	
11	The material uses visual cues (e.g., arrows, boxes, bullets, bold, larger font, highlighting) to draw attention to key points	145	90.6	15	9.4	-	-	91.2	0.49	0.27	0.70	0.89	0.81	0.98			
TOPIC: USE OF VISUAL AIDS																	
14	The material uses visual aids whenever they could make content more easily understood (e.g., illustration of healthy portion size)	154	96.3	6	3.8	-	-	97.5	0.65	0.43	0.87	0.97	0.93	1.01			
15	The material’s visual aids reinforce rather than distract from the content	119	74.4	41	25.6	0	0.0	83.8	0.57	0.35	0.79	0.93	0.90	0.97			
16	The material’s visual aids have clear titles or captions	91	56.9	69	43.1	0	0.0	83.8	0.67	0.45	0.89	0.92	0.88	0.96			
17	The material uses illustrations and photographs that are clear and uncluttered	128	80.0	31	19.4	1	0.6	80.0	0.38	0.17	0.59	0.93	0.89	0.96			
18	The material uses simple tables with short and clear row and column headings	43	26.9	52	32.5	65	40.6	77.5	0.66	0.50	0.81	0.75	0.62	0.96			
**ACTIONABILITY**								**85.4**	**0.75**	**0.69**	**0.81**	**0.86**	**0.82**	**0.90**	**0.8**	**0.63**	**0.88**
19	The material clearly identifies at least one action the user can take	158	98.8	2	1.3	-	-	100.0	1.00	0.78	1.22	1.00	1.00	1.00			
20	The material addresses the user directly when describing actions	119	74.4	41	25.6	-	-	78.8	0.44	0.22	0.66	0.66	0.49	0.82			
21	The material breaks down any action into explicit steps	106	66.3	54	33.8	-	-	85.0	0.66	0.45	0.88	0.73	0.58	0.88			
22	The material provides a tangible tool (e.g., menu planners, checklists) whenever it could help the user take action	85	53.1	75	46.9	0	0.0	78.8	0.57	0.35	0.79	0.89	0.85	0.94			
23	The material provides simple instructions or examples of how to perform calculations	17	10.6	19	11.9	124	77.5	92.5	0.80	0.63	0.97	0.93	0.85	1.00			
24	The material explains how to use the charts, graphs, tables, or diagrams to take actions	40	25.0	39	24.4	81	50.6	80.0	0.68	0.52	0.84	0.57	0.35	0.79			
25	The material uses visual aids whenever they could make it easier to act on the instructions	94	58.8	66	41.3	-	-	82.5	0.64	0.42	0.86	0.66	0.49	0.83			

We used kappa coefficients for the evaluation of each item rather than the total score. For understandability and actionability scores, we calculated Fleiss’s kappa and ICC.

**Table 2 ijerph-19-15763-t002:** Item-level reliability and validity results for items in the Japanese PEMAT-A/V.

Item #	Item	Score Results (Sum of Two Raters)						Inter-Rater Reliability									
		1	(%)	0	(%)	N/A	(%)	% Agree	Fleiss’s κ	(95% CI)		Gwet’s AC1	(95% CI)		ICC	(95% CI)	
**UNDERSTANDABILITY**								**84.3**	**0.72**	**0.65**	**0.79**	**0.90**	**0.88**	**0.93**	**0.76**	**0.65**	**0.84**
TOPIC: CONTENT																	
1	The material makes its purpose completely evident from the beginning	44	48.9	46	51.1	-		82.2	0.64	0.35	0.94	0.64	0.41	0.87			
TOPIC: WORD CHOICE & STYLE																	
3	The material uses common, everyday language	34	37.8	56	62.2	-		82.2	0.62	0.33	0.91	0.66	0.44	0.89			
4	When used, medical terms are clearly defined	27	30.0	63	70.0	-		93.3	0.84	0.55	1.13	0.89	0.75	1.02			
TOPIC: ORGANIZATION																	
7	The material breaks or “chunks” information into short sections	66	73.3	24	26.7	0	0.0	86.7	0.66	0.37	0.95	0.95	0.90	0.99			
8	The material’s sections have informative headers	51	56.7	39	43.3	0	0.0	88.9	0.77	0.48	1.07	0.95	0.90	0.99			
9	The material presents information in a logical sequence	65	72.2	25	27.8	0	0.0	71.1	0.28	-0.01	0.57	0.88	0.81	0.95			
10	The material provides a summary	30	33.3	60	66.7	0	0.0	86.7	0.70	0.41	0.99	0.94	0.89	0.99			
TOPIC: LAYOUT & DESIGN																	
11	The material uses visual cues (e.g., arrows, boxes, bullets, bold, larger font, highlighting) to draw attention to key points	66	73.3	23	25.6	1	1.1	84.4	0.61	0.33	0.89	0.91	0.82	1.00			
12	Text on the screen is easy to read	66	73.3	9	10.0	15	16.7	91.1	0.79	0.57	1.02	0.96	0.92	1.00			
13	The material allows the user to hear the words clearly	71	78.9	16	17.8	3	3.3	84.4	0.55	0.29	0.80	0.94	0.89	0.99			
TOPIC: USE OF VISUAL AIDS																	
17	The material uses illustrations and photographs that are clear and uncluttered	65	72.2	17	18.9	8	8.9	80.0	0.54	0.31	0.77	0.91	0.85	0.97			
18	The material uses simple tables with short and clear row and column headings	15	16.7	12	13.3	63	70.0	80.0	0.57	0.35	0.79	0.84	0.70	0.98			
**ACTIONABILITY**								**90.0**	**0.84**	**0.74**	**0.95**	**0.92**	**0.88**	**0.96**	**0.80**	**0.63**	**0.88**
19	The material clearly identifies at least one action the user can take	78	86.7	12	13.3	-		95.6	0.81	0.52	1.10	0.94	0.86	1.03			
20	The material addresses the user directly when describing actions	55	61.1	35	38.9	-		84.4	0.67	0.38	0.96	0.70	0.49	0.92			
21	The material breaks down any action into explicit steps	35	38.9	55	61.1	-		84.4	0.67	0.38	0.96	0.70	0.49	0.92			
24	The material explains how to use the charts, graphs, tables, or diagrams to take actions	5	5.6	6	6.7	79	87.8	95.6	0.80	0.57	1.03	0.96	0.90	1.03			

We used kappa coefficients for the evaluation of each item rather than the total score. For understandability and actionability scores, we calculated Fleiss’s kappa and ICC.

### 3.2. Inter-Rater Reliability

The median (IQR) of the PEMAT score of the printed materials was 76.7% (62.5–87.5%) for understandability and 70.0% (45.0–91.7%) for actionability. The median (IQR) of the audiovisual materials was 63.6% (45.8–75.0%) for understandability and 66.7% (33.3–83.3%) for actionability. The Japanese version of PEMAT showed strong inter-rater reliability. For PEMAT-P, agreement was 87.3%, average Fleiss’s kappa was 0.73, and Gwet’s AC1 was 0.83. For PEMAT-A/V, agreement was 85.7%, average Fleiss’s kappa was 0.78, and Gwet’s AC1 was 0.80.

The kappa range for the understandability items was 0.30–0.84 in PEMAT-P and 0.35–0.84 in PEMAT-A/V. For the actionability items, scores were 0.47–1.00 for PEMAT-P and 0.67–0.81 for PEMAT-A/V. Gwet’s AC1 revealed strong agreement for both scales and material types. In both the PEMAT-P and PEMAT-A/V, understandability and actionability scores showed substantial to perfect reliability (ICC > 0.7) (Table 1 and Table 2).

To examine the strength of influence of missing items (i.e., items that the evaluator scored as N/A) in the overall PEMAT evaluation, we prepared two scenarios for each missing item: a “best-case scenario” (i.e., “agree = 1” for the missing item) and “worst-case” scenario (i.e., “disagree = 0” for the missing item). The scores for the best- and worst-case scenarios were compared with the scores obtained by the original PEMAT scoring method. In addition, we calculated correlation coefficients between the ratings obtained by the original PEMAT scoring method and the best-case scenario scores and between ratings using the original PEMAT scoring method and the worst-case scenario scores. The distribution of scores across scenarios was nearly identical, confirming that this was a robust approach to manage missing items (Appendix A).

### 3.3. Comparison of the PEMAT Understandability Scores and jReadability

We calculated an average readability score to examine the correlation with the PEMAT understandability scores. The average readability score was 2.7 (range 1.1–4.4) for PEMAT-P and 2.8 (range 0.8–4.0) for PEMAT-A/V. These scores indicate that the materials were at the ‘upper intermediate’ level, which can be understood by people who comprehend the language of daily life and some technical terms. There was a moderate positive correlation between the understandability scores and the readability score for printable materials (Pearson’s r = 0.46; 95% CI, 0.27–0.62), and a weak positive correlation for audiovisual materials (Pearson’s r = 0.33; 95% CI, 0.03–0.57).

### 3.4. Testing with the General Public

#### 3.4.1. Baseline Participant Characteristics

Participant recruitment and surveys were conducted from 18 to 22 June 2021. For PEMAT-P, out of 1526 randomized participants, we analyzed 400 in the high-scoring material group and 399 in the low-scoring material group. For PEMAT-A/V, of 1211 participants randomized, 400 in the high-scoring material group and 400 in the low-scoring material group were analyzed (Figure 1). Those who incorrectly answered the screening questions or did not complete the questionnaire were removed from the analysis. Study arm characteristics are described in Table 3 and Table 4.

#### 3.4.2. Assessment of the Material by Non-Experts

In both PEMAT-P and PEMAT-AV, the high-scoring material group had significantly higher scores for understandability and actionability than the low-scoring material group (PEMAT-P: overall understandability 6.53 vs. 5.96, *p* < 0.001; overall actionability 6.04 vs. 5.49, *p* < 0.001; PEMAT-A/V: overall understandability 7.65 vs. 6.76, *p* < 0.001; overall actionability 7.40 vs. 6.36, *p* < 0.001). Similar results were obtained for all the selected PEMAT items (Table 5).

#### 3.4.3. Self-Efficacy

In PEMAT-AV, perceived self-efficacy significantly increased in the high-scoring material group than in the low-scoring material group (increase in self-efficacy scores 2.18 vs. 1.46, *p* < 0.01). In PEMAT-P, the scores increased more in the high-scoring material group than the low-scoring material group; however, the difference did not reach significance (increase in self-efficacy scores 2.22 vs. 1.53, *p* = 0.14). (Table 6).

## 4. Discussion

### 4.1. Main Findings

We developed and tested the reliability and validity of the Japanese version of PEMAT, a tool for assessing the understandability and actionability of patient education materials. The inter-rater reliability was moderate when measured by the kappa coefficient but showed a convincingly strong agreement when calculated with Gwet’s AC1. In the development of the original PEMAT, the overall agreement was 69–90%, and Gwet’s AC1 range was 0.56–0.86 (mean 0.74) [4]. In the reliability testing of the Malay version of PEMAT, Wong et al. evaluated 13 leaflets and 13 videos. They found that understandability of PEMAT-P had an agreement of 61.5–91.6%, and Gwet’s AC1 was 0.26–0.97. For actionability, agreement was 69.7–98.3% and Gwet’s AC1 was 0.394–0.980. For PEMAT-A/V, the agreement was 64.1–98.3% and Gwet’s AC1 was 0.40–0.98 for understandability. The agreement was 79.5–91.5%, and the AC1 statistic was 0.40–0.93 for actionability [8]. Our study demonstrated that the reliability of the Japanese version of PEMAT is not considerably different from other language versions. In addition, when testing inter-rater reliability, our study included material on diverse topics: from recommendations for healthy living (having medical checkups, improving dietary and exercise habits) to secondary/tertiary prevention in patients with diabetes and cardiovascular disease. This suggests that the Japanese version of PEMAT can evaluate a wide range of health materials in the real world.

In the validation testing with the non-experts, the high-scoring material group rated higher than the low-scoring material group on the PEMAT-P and PEMAT-AV for all eight selected items. This result suggests that materials that medical professionals rated as easy to understand and act upon were validated using the Japanese version of PEMAT. At the time of the development of the original version of PEMAT, there were significant positive correlations between PEMAT-A/V actionability scores and consumer actionability scores [4]. However, there was no clear relationship between understandability as rated by the experts and non-experts’ comprehension scores [4]. This may be attributed to the inadequate sample size of *n* = 47 for consumer testing. In our study, we were able to secure a sufficiently large sample size of nearly 800 participants for each of PEMAT-P and A/V and overcome the limitations of the original version. In addition, in the Japanese version of PEMAT, the increase in self-efficacy tended to be greater in the high-scoring material group than in the low-scoring material group. As self-efficacy is a predictor of behavior [23], materials that receive high ratings on the PEMAT may encourage individuals’ behaviors. Studies have shown that health materials that are easy to understand and act upon may encourage their audiences to adopt healthier behaviors. Arterburn et al. found that understandable decision aids improved the quality of decision-making and reduced uncertainty about the treatment for bariatric surgery in obese patients [30]. Nagle et al. reported pregnant women who viewed a decision aid for prenatal testing of fetal abnormalities were more likely to make an informed decision than those who viewed the less informative material [31].

Nakayama et al. note that 85.4% of Japanese people have inadequate health literacy [32]. However, according to Yamamoto et al., the drug guides for patients written in Japanese are designed to be understandable for patients with at least a high school education level [33]. Thus, it is essential to create and improve materials so they are easy to understand and act on, regardless of individuals’ health literacy. It is also important to improve the understandability and actionability of patient materials in order to communicate the findings of epidemiological and clinical studies to patients and improve patient outcomes. Evaluating and improving materials using the Japanese version of PEMAT may contribute to supporting behavior change in terms of health literacy.

### 4.2. Limitations of This Study

There are limitations to our study. First, although we did not use quantitative measures such as the content validity ratio (CVR) and content validity index (CVI) to evaluate the content validity, we followed the COSMIN methodology used to assess the content validity of PROMs [34] in the expert panel meeting. Second, the validation survey indicated whether the material was easy to understand, but did not measure whether individuals actually understood the information. This was due to the lack of novelty in the materials on eating behavior and exercise, making it impossible to create a comprehension test that would specifically tap knowledge of the content of the materials. Therefore, it is desirable to have non-healthcare professionals quantitatively evaluate materials on a specific disease to measure comprehension and understandability in PEMAT. Third, we performed two rounds of inter-rater reliability evaluation. The modifications made to the User’s Guide between the first and second rounds were important to increase the reliability and usability of the scale. In addition, we could not assess concurrent validity because we could not find validated tools that were similar to PEMAT available in Japanese. However, in the domain of understandability, we observed a moderate positive correlation with jReadability, which has criterion-related validity with the Japanese Language Proficiency Test, supporting the comprehensibility of the Japanese version of PEMAT. Lastly, although we assessed perceived self-efficacy, we could not measure actual behavior. Future research is needed to measure actual outcomes in terms of behavioral change by following up with participants for some time after the intervention.

## 5. Conclusions

The Japanese version of PEMAT developed in this study is the first reliability-validated tool for assessing the patient-friendliness of patient education materials created in Japanese. This instrument indicated that, as with the original version of PEMAT, the materials that experts rated as easy to understand and act upon using the Japanese version of PEMAT were also easy for laypeople to understand and act on. The Japanese version of PEMAT enables medical professionals to select more understandable and actionable patient education materials. It also allows for them to develop and improve patient-friendly materials, ultimately encouraging patients to practice self-management and healthy behaviors. (The Japanese version of the PEMAT is available free of charge for use in non-commercial projects.)

## Figures and Tables

**Figure 1 ijerph-19-15763-f001:**
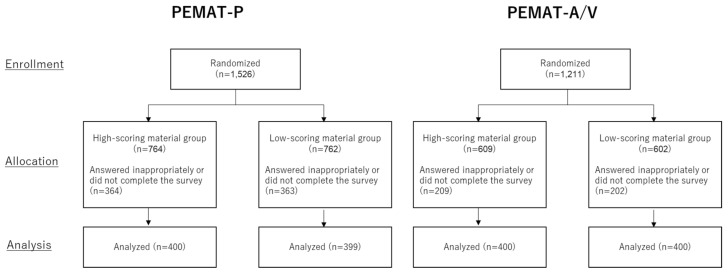
Flow diagram of the participants.

**Table 3 ijerph-19-15763-t003:** Demographic characteristics of participants (PEMAT-P).

Characteristics	High-Scoring Material (*n* = 400)		Low-Scoring Material (*n* = 400)	
Gender (male), *n* (%)	244	61.0	223	55.9
Age, mean (SD)	46.0	11.6	46.6	12.1
BMI, mean (SD)	22.5	3.7	22.4	3.7
Education, *n* (%)				
Less than high school	5	1.3	2	0.5
High school graduate	116	29.0	111	27.8
Some college	78	19.5	83	20.8
College graduate	180	45.0	170	42.6
Graduate school	21	5.3	23	5.8
Occupation, *n* (%)				
Office worker (regular employee)	172	43.0	186	46.6
Office worker(contract employee)	30	7.5	20	5.0
Public officer	17	4.3	24	6.0
Self-employed	26	6.5	23	5.8
Manager/executive officer	12	3.0	8	2.0
Part-time worker	57	14.3	57	14.3
Student	5	1.3	9	2.3
Homemaker	44	11.0	38	9.5
Unemployed	37	9.3	34	8.5
Annual household income, *n* (%)				
Less than 30,000 USD	79	19.8	84	21.1
30,000 to 50,000 USD	119	29.8	119	29.8
50,000 to 70,000 USD	80	20.0	66	16.5
70,000 to 100,000 USD	72	18.0	88	22.1
100,000 to 150,000 USD	38	9.5	33	8.3
More than 150,000 USD	12	3.0	9	2.3
Marital status, *n* (%)				
Single	128	32.0	124	31.1
Married	248	62.0	250	62.7
Divorced or bereaved	24	6.0	25	6.3
Self-perceived health status, *n* (%)				
Very good	30	7.5	23	5.8
Good	220	55.0	214	53.6
Fair	79	19.8	78	19.5
Bad	53	13.3	67	16.8
Very bad	18	4.5	17	4.3
Health literacy (HLS-14), mean (SD)	52.1	7.8	52.8	7.8
Self-efficacy, mean (SD)	1.7	10.2	2.2	10.6

BMI = body mass index, HLS-14 = Health Literacy Scale-14. *t*-test: age, BMI, health literacy (HLS-14), self-efficacy. chi-square test: gender, Fisher’s exact test: others.

**Table 4 ijerph-19-15763-t004:** Demographic characteristics of participants (PEMAT-A/V).

Characteristics	High-Scoring Material (*n* = 400)		Low-Scoring Material (*n* = 400)	
Gender (male), *n* (%)	311	77.8	312	78.0
Age, mean (SD)	66.05	4.39	66.37	4.99
Education, *n* (%)				
Less than high school	5	1.3	9	2.3
High school graduate	103	25.8	118	29.5
Some college	65	16.3	54	13.5
College graduate	217	54.3	200	50.0
Graduate school	10	2.5	19	4.8
Occupation, *n* (%)				
Office worker (regular employee)	50	12.5	58	14.5
Office worker(contract employee)	33	8.3	33	8.3
Public officer	15	3.8	11	2.8
Self-employed	51	12.8	56	14.0
Manager/executive officer	23	5.8	14	3.5
Part-time worker	65	16.3	48	12.0
Student	0	0.0	0	0.0
Homemaker	41	10.3	46	11.5
Unemployed	122	30.5	134	33.5
Annual household income, *n* (%)				
Less than 30,000 USD	121	30.3	152	38.0
30,000 to 50,000 USD	141	35.3	119	29.8
50,000 to 70,000 USD	62	15.5	49	12.3
70,000 to 100,000 USD	43	10.8	51	12.8
100,000 to 150,000 USD	21	5.3	19	4.8
More than 150,000 USD	12	3.0	10	2.5
Marital status, *n* (%)				
Single	36	9.0	40	10.0
Married	316	79.0	316	79.0
Divorced or bereaved	48	12.0	44	11.0
Self-perceived health status, *n* (%)				
Very good	20	5.0	14	3.5
Good	231	57.8	214	53.5
Fair	86	21.5	83	20.8
Bad	54	13.5	74	18.5
Very bad	9	2.3	15	3.8
Physical activity, *n* (%)				
Vigorous	11	2.8	11	2.8
Moderate	119	29.8	123	30.8
Light	270	67.5	266	66.5
Health literacy (HLS-14), mean (SD)	51.9	7.9	51.6	8.1
Self-efficacy, mean (SD)	11.7	4.3	12.0	4.5

BMI = body mass index, HLS-14 = Health Literacy Scale-14. *t*-test: age, BMI, health literacy (HLS-14), self-efficacy. chi-square test: gender, Fisher’s exact test: others.

**Table 5 ijerph-19-15763-t005:** Assessment of the material by non-experts.

PEMAT-P					
**Item**	**High-Scoring Material**		**Low-Scoring Material**		***p*-Value**
	**Score**	**SD**	**Score**	**SD**	
**Overall understandability**	6.53	2.32	5.96	2.08	<0.001
The material makes its purpose completely evident. (Item 1)	6.24	2.23	5.89	2.11	0.024
The material uses common, everyday language. (Item 3)	6.65	2.19	6.2	2.07	0.003
The material’s sections have informative headers. (Item 8)	6.48	2.2	6.09	2.14	0.012
The material presents information in a logical sequence. (Item 9)	6.28	2.15	5.92	1.98	0.015
The material uses visual cues to draw attention to key points. (Item 11)	6.3	2.15	5.87	2.1	0.004
The material uses clear illustrations and photographs. (Item 17)	6.5	2.24	5.89	2.07	<0.001
**Overall actionability**	6.04	2.17	5.49	1.93	<0.001
The material clearly identifies at least one action the user can take. (Item 19)	6.24	2.16	5.74	1.95	0.001
The material breaks down any action into manageable, explicit steps. (Item 21)	6.08	2.13	5.57	1.86	<0.001
PEMAT-A/V					
**Item**	**High-Scoring Material**		**Low-Scoring Material**		***p*-Value**
	**Score**	**SD**	**Score**	**SD**	
**Overall understandability**	7.65	2.17	6.76	2.12	<0.001
The material makes its purpose completely evident. (Item 1)	7.54	2.21	7.08	2.23	0.003
The material uses common, everyday language. (Item 3)	7.64	2.18	6.7	2.16	<0.001
The material’s sections have informative headers. (Item 8)	7.44	2.13	6.56	2.17	<0.001
The material presents information in a logical sequence. (Item 9)	7.53	2.19	6.9	2.1	<0.001
The material uses visual cues to draw attention to key points. (Item 11)	7.23	2.15	6.64	2.11	<0.001
The material uses clear illustrations and photographs. (Item 17)	7.2	2.13	6.72	2.07	0.001
**Overall actionability**	7.48	2.22	6.36	2.11	<0.001
The material clearly identifies at least one action the user can take. (Item 19)	7.6	2.17	6.58	2.11	<0.001
The material breaks down any action into manageable, explicit steps. (Item 21)	7.5	2.15	6.3	2.1	<0.001

*t*-test, no directional assumption; SD = standard deviation.

**Table 6 ijerph-19-15763-t006:** Perceived self-efficacy of the participants.

PEMAT-P					
	**High-Scoring Material**		**Low-Scoring Material**		***p*-Value**
	**Mean**	**SD**	**Mean**	**SD**	
Baseline	1.71	10.16	2.22	10.55	0.486
Post-intervention	3.94	11.07	3.75	10.84	0.811
Difference	2.22	7.09	1.53	6.28	0.142
PEMAT-A/V					
	**High-Scoring Material**		**Low-Scoring Material**		***p*-Value**
	**Mean**	**SD**	**Mean**	**SD**	
Baseline	11.74	4.34	12.04	4.49	0.345
Post-intervention	13.92	4.81	13.50	4.86	0.217
Difference	2.18	3.76	1.46	3.20	0.004

*t*-test, no directional assumption; SD = standard deviation.

## Data Availability

The data presented in this study are available on request from the corresponding author. The data are not publicly available due to privacy.

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
