# Peer review of "Translation, Cross-Cultural Adaptation, and Validation of the Japanese Version of the Patient Education Materials Assessment Tool (PEMAT)"

_ijerph, 2022, doi:10.3390/ijerph192315763_

Round 1

Reviewer 1 Report

The manuscript entitled “Translation, Cross-Cultural Adaptation, and Validation of the Japanese Version of The Patient Education Materials Assessment Tool (PEMAT)” contains an interesting and valuable study on developing a version of PERMAT for the Japanese public.  Although the study is methodologically sound, a few issues may need to be addressed before publication is warranted.

The introductory section may benefit from a broader coverage of studies that have attempted to introduce a measurement tool to a country whose citizens speak a language and possess a culture that may be substantially different from the language and culture of the sample used to develop the measurement tool in the first place. In general, what kind of guidance such studies have offered to the authors of the current study?

At the start of the method section, a brief paragraph should summarize the steps taken by the authors to assess the validity and reliability of the Japanese version of PERMAT. Such a section will help the reader to better understand the steps taken by the authors away from the details of implementation.

The content validity section is missing critical information. In my modest opinion, Stage 2 (Assessment of Content Validity by the Expert Panel) needs to contain an accurate description of (a) the selection of participants, (b) the decisions made by the panel, and (c) their rationale.  Perhaps, some of the information provided in the result section, should be stated earlier to clarify questions that the reader may have (e.g., how was “understandability” operationally defined for PERMAT?  How was “actionability” operationally defined for PERMAT?). Furthermore, what was the feedback provided by the questionnaire's original developers? Was there disagreement between the panel of experts and the developers? If so, how was the disagreement resolved?

The authors state that “[t]he evaluators for this study comprised four experts: two physicians with more than 5 years of clinical experience (EF and RS); one dietitian without clinical experience (RI); 101 and one health communication specialist with an educational background (RY).” What were the criteria used to select evaluators for the measurement of the reliability of the instrument? Were the people serving as evaluators some of the experts who participated in the determination of its content validity (i.e., the panel of 12 experts)?  

 The recruitment of participants from the general public needs to be clarified. The authors state that “[s]tudy participants were recruited from registered monitors of an online survey company (Rakuten Insight).” What was the size of the population relative to the sample of participants? If a large number of individuals did not participate or failed to complete the entire survey, was there a way to assess whether they differed from the participants? Namely, would it be possible to determine whether the sample is representative of the population of interest?

In the discussion section, the authors state that “[o]ur study demonstrated that the reliability of the Japanese version of PEMAT compares favorably with other language versions.” Can the authors provide more information regarding this critical issue? 

Author Response

The manuscript entitled "Translation, Cross-Cultural Adaptation, and Validation of the Japanese Version of The Patient Education Materials Assessment Tool (PEMAT)" contains an interesting and valuable study on developing a version of PERMAT for the Japanese public. Although the study is methodologically sound, a few issues may need to be addressed before publication is warranted.

  1. The introductory section may benefit from a broader coverage of studies that have attempted to introduce a measurement tool to a country whose citizens speak a language and possess a culture that may be substantially different from the language and culture of the sample used to develop the measurement tool in the first place. In general, what kind of guidance such studies have offered to the authors of the current study?
  • In developing the Japanese version of PEMAT, we referred to the development study of the Korean version of PEMAT[1]. Lee et al. created a cardiovascular disease prevention material for Korean immigrants based on the Korean version of PEMAT, and found that they could improve the understandability and actionability of the material[1]. To date, the Korean version is the only reported case study of the use of PEMAT in other languages. However, we believe it is essential for PEMAT to be deployed in multiple languages and for findings on understandability and actionability to be accumulated.
  • We have added sentences in the Introduction (page 2, line 59-63).

--------------------------------------------------------------------

PEMAT has been translated into Malay [2, 3] and Korean [1], but a Japanese version has not yet been developed. Lee et al. created a cardiovascular disease prevention material for Korean immigrants based on the Korean version of PEMAT and found that they could improve the understandability and actionability of the material[1]. Therefore, we believe it is essential for PEMAT to be deployed in multiple languages and for findings on understandability and actionability to be accumulated. Furthermore, to our knowledge, there are no validated tools to assess whether material is understandable and actionable in Japanese.

--------------------------------------------------------------------

At the start of the method section, a brief paragraph should summarize the steps taken by the authors to assess the validity and reliability of the Japanese version of PERMAT. Such a section will help the reader to better understand the steps taken by the authors away from the details of implementation.

  • We have added sentences as follows in the Materials and methods (page 2, lines 68-71).

--------------------------------------------------------------------

  1. Materials and methods
    The development of the Japanese version of the PEMAT consists of five steps. We translated the PEMAT in Step 1, examined content validity in Step 2, examined inter-rater reliability in Step 3, examined convergent validity in Step 4, and examined predictive validity in Step 5.

2.1. Stage 1: Translation of PEMAT into Japanese

--------------------------------------------------------------------

  1. The content validity section is missing critical information. In my modest opinion, Stage 2 (Assessment of Content Validity by the Expert Panel) needs to contain an accurate description of (a) the selection of participants, (b) the decisions made by the panel, and (c) their rationale.
  • (a) the selection of participants

According to the User's Guide of the original PEMAT, PEMAT is intended for use by health professionals,  health librarians, and other professionals who provide health and medical information to patients and the general public [4]. In this study, we recruited experts with the attributes of potential PEMAT users for the content validity evaluation. In Japan, the creators of materials for patients are not limited to medical professionals or those with clinical experience; non-medical professionals from the government, health insurance, and various for-profit companies may also be involved in creating materials. Based on this current situation, we selected experts with the following attributes: medical specialists with clinical experience (two doctors, one nurse, one pharmacist, and two dieticians); non-medical specialists (two patient advocates, one editor with experience in developing health and medical materials, two media professionals, and one health communication researcher with a background in education.) 

  • (b) the decisions made by the panel, and (c) their rationale

At the expert panel meeting, the content validation was evaluated based on the following three aspects: 1) relevance (whether the items were relevant to the constructs of interest in the particular population and context of use), 2) comprehensiveness (whether essential aspects of the construct were missing), and 3) comprehensibility (whether the items were understood by the raters as intended). The decision by the panel and the rationale for the decision are added to the Materials and methods and Table S1.

  • We added sentences as follows in the Materials and methods (page 2, line 87-107).

--------------------------------------------------------------------

At a panel meeting of the experts (medical professionals and health communication researchers), we determined whether the items apply to the Japanese cultural background and whether the content is appropriate. The original PEMAT is intended for use by health professionals, health librarians, and other professionals who provide health and medical information to patients and the general public[4, 5]. Following this, our study recruited experts with the attributes of potential PEMAT users for the content validity evaluation. The expert panel consisted of twelve members: medical specialists with clinical experience (two doctors, one nurse, one pharmacist, and two dieticians); non-medical specialists (two patient advocates, one editor with experience in developing health and medical materials, two media professionals, and one health communication researcher with a background in education.) At the meeting, content validity was assessed based on: 1) relevance (whether the items were relevant to the constructs of interest in the particular population and context of use), 2) comprehensiveness (whether essential aspects of the construct were missing), and 3) comprehensibility (whether the items were understood by the raters as intended). When there were discrepancies among the panel of experts, we consulted with a third person (TK) for reconciliation. If issues or questions were not resolved at the meeting, we asked the developer of the original PEMAT (CB) for feedback.

--------------------------------------------------------------------

Perhaps, some of the information provided in the result section, should be stated earlier to clarify questions that the reader may have (e.g., how was "understandability" operationally defined for PEMAT? How was "actionability" operationally defined for PEMAT?).

  • We introduced the following definitions of understandability and actionability in PEMAT when we described the original version of PEMAT to our readers in the Introduction (page 1, line 42-43)

--------------------------------------------------------------------

Understandability refers to the extent to which consumers of diverse backgrounds can process and explain key messages [5]. Actionability refers to the degree to which consumers of diverse backgrounds and varying levels of health literacy can identify what actions they should take to improve their health, based on the information presented [5].

--------------------------------------------------------------------

Furthermore, what was the feedback provided by the questionnaire's original developers? Was there disagreement between the panel of experts and the developers? If so, how was the disagreement resolved?

When there were disagreements among the panel of experts, we conducted a third-person (TK) reconciliation. If the issues or questions were not resolved at the panel meeting, we consulted the developer of the original version of PEMAT (CB). We add the feedback from the developer to Table S1.

  • We added sentences as follows in the Materials and methods (page 2, line 87-107).

--------------------------------------------------------------------

At a panel meeting of the experts (medical professionals and health communication researchers), we determined whether the items apply to the Japanese cultural background and whether the content is appropriate. The original PEMAT is intended for use by health professionals, health librarians, and other professionals who provide health and medical information to patients and the general public[4, 5]. Following this, our study recruited experts with the attributes of potential PEMAT users for the content validity evaluation. The expert panel consisted of twelve members: medical specialists with clinical experience (two doctors, one nurse, one pharmacist, and two dieticians); non-medical specialists (two patient advocates, one editor with experience in developing health and medical materials, two media professionals, and one health communication researcher with a background in education.) At the meeting, content validity was assessed based on: 1) relevance (whether the items were relevant to the constructs of interest in the particular population and context of use), 2) comprehensiveness (whether essential aspects of the construct were missing), and 3) comprehensibility (whether the items were understood by the raters as intended). When there were discrepancies among the panel of experts, we consulted with a third person (TK) for reconciliation. If issues or questions were not resolved at the meeting, we asked the developer of the original PEMAT (CB) for feedback.

--------------------------------------------------------------------

  • We added the decision by the panel, the rationale for the decision, and the developer's feedback to Table S1.

The authors state that "[t]he evaluators for this study comprised four experts: two physicians with more than 5 years of clinical experience (EF and RS); one dietitian without clinical experience (RI); 101 and one health communication specialist with an educational background (RY)." What were the criteria used to select evaluators for the measurement of the reliability of the instrument? Were the people serving as evaluators some of the experts who participated in the determination of its content validity (i.e., the panel of 12 experts)?  

  • We recruited our raters from a subset of our panel of content validity meetings with consideration for human resources. The original version of the PEMAT was already a well-established tool, and we, too, did not make any major changes in the Japanese version of the content validation except for deleting of item 5 from the original version. Therefore, we believe that participation in content validity meetings would not broadly impact inter-rater reliability.

The recruitment of participants from the general public needs to be clarified. The authors state that "[s]tudy participants were recruited from registered monitors of an online survey company (Rakuten Insight)." What was the size of the population relative to the sample of participants? If a large number of individuals did not participate or failed to complete the entire survey, was there a way to assess whether they differed from the participants? Namely, would it be possible to determine whether the sample is representative of the population of interest?

  • Rakuten Insight reported that approximately 2.2 million active monitors (who have logged into their registered accounts within 12 months) registered in the panel as of September 2022[6]. Since we recruited from among all registered monitors in this study who met the age range for inclusion, there was a certain degree of possibility that our sample was representative of the population (general public).
  • We have noted to Materials and methods regarding the survey panels and the recruitment of participants for this study (page 4, line 175-181).

--------------------------------------------------------------------

2.5.1 Participants
Study participants were recruited from registered monitors of an online survey company (Rakuten Insight). The survey company reported that approximately 2.2 million active monitors (who have logged into their registered accounts within 12 months) registered in the panel as of September 2022[6]. Men and women who use Japanese as a native language were eligible to participate in the study. We solicited the monitors who met the age criteria described below to participate in the survey via email or push notification via the survey company and conducted a screening survey of all who agreed to participate.
--------------------------------------------------------------------

In the discussion section, the authors state that "[o]ur study demonstrated that the reliability of the Japanese version of PEMAT compares favorably with other language versions." Can the authors provide more information regarding this critical issue? 

  • We have included the following statements about reliability coefficients for other language versions in the 1 Main findings part of the Discussion.
  • We restated certain parts of the description and modified some of the notations (page 12, lines 387-396).

--------------------------------------------------------------------

In the development of the original PEMAT, the overall agreement was 69–90%, and Gwet's AC1 range was 0.56–0.86 (mean 0.74) [5]. In the reliability testing of the Malay version of PEMAT, Wong et al. evaluated 13 leaflets and 13 videos. They found that understandability of PEMAT-P had an agreement of 61.5–91.6%, and Gwet's AC1 was 0.26–0.97. For actionability, agreement was 69.7–98.3% and Gwet's AC1 was 0.394–0.980. For PEMAT-A/V, the agreement was 64.1–98.3% and Gwet's AC1 was 0.40–0.98 for understandability. The agreement was 79.5–91.5%, and the AC1 statistic was 0.40–0.93 for actionability [2]. Our study demonstrated that the reliability of the Japanese version of PEMAT is not considerably different from other language versions.

--------------------------------------------------------------------

References

  1. Lee, H., et al., Development and Evaluation of Cardiovascular Disease Prevention Education Materials for Middle-aged Korean-Chinese Female Workers: Applying Patient Education Materials Assessment Tool for Printable Materials (PEMAT-P). Journal of Korean Academy of Community Health Nursing, 2016. 27(3): p. 284.
  2. Wong, S.T., N. Saddki, and W.N. Arifin, Inter-Rater Reliability of the Bahasa Malaysia Version of Patient Education Materials Assessment Tool. The Medical journal of Malaysia, 2019. 74: p. 100.
  3. Wong, S.T., N. Saddki, and W.N. Arifin, Validity of the Bahasa Malaysia Version of Patient Education Materials Assessment Tool. Malaysian Journal of Public Health Medicine, 2019. 19: p. 35.
  4. The Patient Education Materials Assessment Tool (PEMAT) and User’s Guide. 2014; Available from: https://www.ahrq.gov/health-literacy/patient-education/pemat.html.
  5. Shoemaker, S.J., M.S. Wolf, and C. Brach, Development of the Patient Education Materials Assessment Tool (PEMAT): a new measure of understandability and actionability for print and audiovisual patient information. Patient Educ Couns, 2014. 96(3): p. 395-403.

Reviewer 2 Report

It was a very careful analysis and examination, and I think it is a good research that can be expected to be used internationally by PEMAT. However, it seems that tables in academic papers have not been created. I think that it is better to carefully create tables by referring to the style manual. Also, please consider the following points in detail.

1. Stage 3: For PEMAT evaluators. What kind of experts are the evaluators that the original PEMAT envisions? Or is it intended for amateurs? Does the evaluator in this study include the evaluator assumed in the original?

2. Stage 4: Why is "jReadability" a target indicator of PEMAT's convergent validity?

3. Stage 5: Predictive validity of PEMAT. A theoretical explanation is necessary before this examination. Can you hypothesize that PEMAT ratings predict readers' self-efficacy? What theoretical basis does PEMAT itself have?It is necessary to clearly explain under what theory PEMAT is set.

4. About Table. Where (%) or (95% CI) is given, the values in the table should also be shown in brackets. SD is better indicated as (SD).

5. In Table3 and Table4, the left column is in one column, but it is very difficult to read. Express variable names and category names in a hierarchical manner using indentation.

6. Asterisks are not required when P-values are indicated. P-values should be 3 digits after the decimal point, and scores less than 0.001 should be indicated with inequality signs.

Author Response

It was a very careful analysis and examination, and I think it is a good research that can be expected to be used internationally by PEMAT. However, it seems that tables in academic papers have not been created. I think that it is better to carefully create tables by referring to the style manual. Also, please consider the following points in detail.

  1. Stage 3: For PEMAT evaluators. What kind of experts are the evaluators that the original PEMAT envisions? Or is it intended for amateurs? Does the evaluator in this study include the evaluator assumed in the original?
  • According to the User's Guide of the original PEMAT, PEMAT is intended for use by health professionals, health librarians, and other professionals who provide health and medical information to patients and the general public [4]. In this study, we recruited experts with the attributes of potential PEMAT users for the evaluation of reliability. In Japan, the creators of materials for patients are not limited to medical professionals or those with clinical experience; non-medical professionals from the government, health insurance, and various for-profit companies may also be involved in creating materials. As described in "2.3. Stage 3: Determining the reliability of the instrument," the evaluators who participated in the reliability assessment for this study consisted of four experts: two physicians with more than five years of clinical experience (EF and RS); one dietitian without clinical experience (RI); and one health communication specialist with an educational background (RY).

  1. Stage 4: Why is "jReadability" a target indicator of PEMAT's convergent validity?
  • To our knowledge, there are no other Japanese tools for evaluating materials that have been tested for reliability and validity other than the PEMAT. In the absence of a gold standard, we used the readability score, which is the closest to understandability in the PEMAT, to test convergent validity. It has been demonstrated that the jReadability formula can predict the difficulty of a text with a high degree of accuracy. Furthermore, the differences in readability can be detected even when analyzing data other than that used to create the readability formulas (the Japanese Language Proficiency Test) [7].
  • We have added the features of jReadability to Materials and methods (page 3, line 145-156).

------------------------------

2.4. Stage 4: Testing convergent validity with readability scores

We evaluated the materials included in Stage 3 with a readability scale to test convergent validity. EF proposed the text readability measurement system "jReadability" [8, 9], which is a web system for automatically evaluating the readability of Japanese text, as the developers of the original PEMAT recommended using readability evaluation tools to evaluate the readability of printed materials in addition to the PEMAT [10]. It has been demonstrated that the jReadability formula can predict the difficulty of a text with a high degree of accuracy. Furthermore, it has been shown that differences in readability can be detected even when analyzing data other than that used to create the readability formulas (the Japanese Language Proficiency Test) [7]. To assess readability,…

-----------------------------

  1. Stage 5: Predictive validity of PEMAT. A theoretical explanation is necessary before this examination. Can you hypothesize that PEMAT ratings predict readers' self-efficacy? What theoretical basis does PEMAT itself have?It is necessary to clearly explain under what theory PEMAT is set.
  • When assessing predictive validity, it is ideal to measure the change in participants' behavior before and after viewing the materials. In this study, we employed Prochaska's transtheoretical model and Bandura's social cognitive theory as a basis for predicting the behavior of the audience of the materials[11-13]. Bandura stated that for behavior change, it is vital to increase self-efficacy, which means confidence in carrying out the behavior and overcoming temptations that prevent change[11, 12]. Self-efficacy is likely to change over a shorter period and improve as the stage of behavioral change progresses. We incorporated self-efficacy as a surrogate outcome variable for participants' behavior, hypothesizing the understandability and actionability scores assessed on the PEMAT predicted self-efficacy.

  • We have added sentences in the Materials and methods (page 5, line 209-220).

--------------------------------------------

2.5.3. Measures

The survey company provided participants' gender and age, and participants responded to questions about their educational background, annual family income, occupation, marital status, and self-perceived health. Participants also answered questions about their baseline health literacy and perceived self-efficacy. Measuring the change in behavior before and after viewing the materials as an outcome would be ideal to verify whether materials rated highly on the PEMAT are more likely to support participants to take action. However, participants may find it difficult to take action (e.g., cooking healthy meals, going out for exercise) immediately after viewing the materials. It is also not feasible to measure behavioral implementation in an online survey. Bandura stated that for behavior change, it is vital to increase self-efficacy, which means confidence in carrying out the behavior and overcoming temptations that prevent change[11, 12]. Self-efficacy is likely to change over a shorter period and improve as the stage of behavioral change progresses. We measured participants’ self-efficacy to examine the predictive validity, hypothesizing the understandability and actionability scores assessed on the PEMAT predicted self-efficacy. Health literacy was measured using the 14-item health literacy scale for Japanese adults (HLS-14) [14]. Self-efficacy was measured by the Self-Efficacy Scale for Positive Eating Behavior [15] for PEMAT-P and the Home-Exercise Barrier Self-Efficacy Scale [16] for PEMAT-A/V.

--------------------------------------------

  1. About Table. Where (%) or (95% CI) is given, the values in the table should also be shown in brackets. SD is better indicated as (SD).
  • We corrected the SD notations in Tables 3 and 4.

  1. In Table3 and Table4, the left column is in one column, but it is very difficult to read. Express variable names and category names in a hierarchical manner using indentation.
  • We have added indents to the variables and categories in Tables 3 and 4.

  1. Asterisks are not required when P-values are indicated. P-values should be 3 digits after the decimal point, and scores less than 0.001 should be indicated with inequality signs.
  • We have corrected the notation of P-values in Tables 5 and 6.

References

  1. Lee, H., et al., Development and Evaluation of Cardiovascular Disease Prevention Education Materials for Middle-aged Korean-Chinese Female Workers: Applying Patient Education Materials Assessment Tool for Printable Materials (PEMAT-P). Journal of Korean Academy of Community Health Nursing, 2016. 27(3): p. 284.
  2. Wong, S.T., N. Saddki, and W.N. Arifin, Inter-Rater Reliability of the Bahasa Malaysia Version of Patient Education Materials Assessment Tool. The Medical journal of Malaysia, 2019. 74: p. 100.
  3. Wong, S.T., N. Saddki, and W.N. Arifin, Validity of the Bahasa Malaysia Version of Patient Education Materials Assessment Tool. Malaysian Journal of Public Health Medicine, 2019. 19: p. 35.
  4. The Patient Education Materials Assessment Tool (PEMAT) and User’s Guide. 2014; Available from: https://www.ahrq.gov/health-literacy/patient-education/pemat.html.
  5. Shoemaker, S.J., M.S. Wolf, and C. Brach, Development of the Patient Education Materials Assessment Tool (PEMAT): a new measure of understandability and actionability for print and audiovisual patient information. Patient Educ Couns, 2014. 96(3): p. 395-403.
  6. RakutenInsight. About the survey panel. 2022; Available from: https://insight.rakuten.co.jp/panel/.
  7. Lee, Readability Research for Japanese Language Education. Waseda studies in Japanese language education, 2016(21): p. 1-16.
  8. Hasebe, Y. and J.-h. Lee, Introducing a readability evaluation system for Japanese language education, in The 6th International Conference on Computer Assisted Systems for Teaching and Learning Japanese [CASTEL/J] 2015: University of Hawaii.
  9. Lee, J.-h. and Y. Hasebe. jReadability PORTAL. 2013 [cited 2021 Jan 7]; Available from: http://jreadability.net.
  10. Davis, T.C., et al., Low literacy impairs comprehension of prescription drug warning labels. J Gen Intern Med, 2006. 21(8): p. 847-51.
  11. Bandura, A., Self-efficacy: toward a unifying theory of behavioral change. Psychol Rev, 1977. 84(2): p. 191-215.
  12. Bandura, A., Self-efficacy., in Encyclopedia of human behavior, I.V.S. Ramachaudran, Editor. 1994, Academic Press. : New York:. p. 71-81.
  13. Prochaska, J.O. and W.F. Velicer, The transtheoretical model of health behavior change. Am J Health Promot, 1997. 12(1): p. 38-48.
  14. Suka, M., et al., The 14-item health literacy scale for Japanese adults (HLS-14). Environ Health Prev Med, 2013. 18(5): p. 407-15.
  15. Moriya, K., An Examination of the Usefulness of Self Efficacy and Social-Support Scales Providing Support for Positive Health Behavior Change. Bulletin of Tenshi College, 2009. 9: p. 1-20.
  16. Arita, N. and K. Takenaka, Development of a Home-Exercise Barrier Self−Efficacy Scale for Elderly People Requiring Support and Care. The Journal of Japanese Physical Therapy Association., 2014. 41(6): p. 338-346.

Round 2

Reviewer 1 Report

The authors have made a reasonable set of suitable changes to their manuscript. As a result, the study described in the manuscript will be likely to be of interest to a broad readership.

Reviewer 2 Report

Nothing in particular.